# High Risk of Sustained Ventricular Arrhythmia Recurrence After Acute Myocarditis

**DOI:** 10.3390/jcm9030848

**Published:** 2020-03-20

**Authors:** Laurent Rosier, Amir Zouaghi, Valentin Barré, Raphaël Martins, Vincent Probst, Eloi Marijon, Nicolas Sadoul, Samuel Chauveau, Antoine Da Costa, Marc Badoz, Michael Peyrol, Jérémie Barraud, Grégoire Massoullie, Romain Eschalier, Madeline Espinosa, François Lesaffre, Rodrigue Garcia, Bruno Degand, Antoine Noël, Jacques Mansourati, Fabrice Extramiana, Vincent Algalarrondo, Hervé Devilliers, Yves Cottin, Estelle Gandjbakhch, Charles Guenancia

**Affiliations:** 1Cardiology Department, Dijon Bourgogne University Hospital, 21000 Dijon, France; lrosier@hotmail.fr (L.R.); yves.cottin@chu-dijon.fr (Y.C.); 2Cardiology Department, Hôpitaux Universitaires Pitié Salpêtrière, APHP, 75013 Paris, France; amir-zouaghi@hotmail.fr (A.Z.); estelle.gandjbakhch@aphp.fr (E.G.); 3Cardiology Department, University Hospital, 35000 Rennes, France; valentin.barre@chu-rennes.fr (V.B.); raphael.martins@chu-rennes.fr (R.M.); 4Institut du thorax, Service de Cardiologie and INSERM 1087, 44000 Nantes, France; vincent.probst@chu-nantes.fr; 5Cardiology Department, European Georges Pompidou Hospital and Paris Descartes University, 75015 Paris, France; eloi.marijon@inserm.fr; 6Cardiology Department, University Hospital, 54511 Nancy, France; 7Cardiology Department, University Hospital Louis Pradel, 69500 Lyon, France; samuelchauveau@hotmail.com; 8Cardiology Department, University Hospital, 42055 Saint-Etienne, France; antoine.dacosta@univ-st-etienne.fr; 9Cardiology Department, University Hospital, 25030 Besançon, France; mbadoz@chu-besancon.fr; 10Aix-Marseille University, Assistance Publique–Hôpitaux de Marseille (APHM), Department of Cardiology, Nord Hospital, 13000 Marseille, France; michael.peyrol@ap-hm.fr (M.P.); jeremie.barraud@ap-hm.fr (J.B.); 11Cardiology Department, CHU Clermont-Ferrand, Clermont-Ferrand, France and Université Clermont Auvergne, CHU Clermont-Ferrand, CNRS, SIGMA Clermont, Institut Pascal, 63000 Clermont-Ferrand, France; gmassoullie@chu-clermontferrand.fr (G.M.); reschalier@chu-clermontferrand.fr (R.E.); 12Cardiology Department, University Hospital, 51100 Reims, France; mespinosa@chu-reims.fr (M.E.); flesaffre@chu-reims.fr (F.L.); 13CHU Poitiers, Centre Cardiovasculaire, 86000 Poitiers, France; rodrigue_garcia@hotmail.fr (R.G.) ; BRUNO.DEGAND@chu-poitiers.fr (B.D.); 14Cardiology Department, University Hospital, 29200 Brest, France; antoine.noel@chu-brest.fr (A.N.); jacques.mansourati@chu-brest.fr (J.M.); 15Department of Cardiology, Bichat Claude Bernard Hospital, University Paris Diderot, 75018 Paris, France; fabrice.extramiana@aphp.fr (F.E.); vincent.algalarrondo@gmail.com (V.A.); 16Internal Medicine 2 Department, Dijon Bourgogne University Hospital, 21000 Dijon, France; herve.devilliers@chu-dijon.fr; 17PEC 2, Univ. Bourgogne Franche–Comté, 21000 Dijon, France

**Keywords:** myocarditis, ventricular tachycardia, ventricular fibrillation, implantable cardioverter defibrillator

## Abstract

Acute myocarditis is associated with cardiac arrhythmia in 25% of cases; a third of these arrhythmias are ventricular tachycardia (VT) or ventricular fibrillation (VF). The implantation of a cardiac defibrillator (ICD) following sustained ventricular arrhythmia remains controversial in these patients. We sought to assess the risk of major arrhythmic ventricular events (MAEs) over time in patients implanted with an ICD following sustained VT/VF in the acute phase of myocarditis compared to those implanted for VT/VF occurring on myocarditis sequelae. Our retrospective observational study included patients implanted with an ICD following VT/VF during acute myocarditis or VT/VF on myocarditis sequelae, from 2007 to 2017, in 15 French university hospitals. Over a median follow-up period of 3 years, MAE occurred in 11 (39%) patients of the acute myocarditis group and 24 (60%) patients of the myocarditis sequelae group. Kaplan–Meier MAE rate estimates at one and three years of follow-up were 19% and 45% in the acute group, and 43% and 64% in the sequelae group. Patients who experienced sustained ventricular arrhythmias during acute myocarditis had a very high risk of VT/VF recurrence during follow-up. These results show that the risk of MAE recurrence remains high after resolution of the acute episode.

## 1. Introduction

Myocarditis is an inflammatory disease of the myocardium that has a wide variety of clinical presentations and whose clinical course is poorly understood. Acute myocarditis is associated with arrhythmias in about one-quarter of patients; one-third of the arrhythmias are either ventricular tachycardia (VT) or ventricular fibrillation (VF) [1,2].

The latest European and American recommendations clearly support the implantation of a cardiac defibrillator (ICD) as secondary prevention in patients who have had a major arrhythmic event (MAE) subsequent to myocarditis. However, implantation following severe ventricular arrhythmia in the acute phase of myocarditis is disputed, and according to some recommendations it should be limited to giant cell myocarditis and cardiac sarcoidosis [3,4]. Clinical approaches are varied: some centers implant patients during the acute phase of myocarditis, regardless of etiology, while other centers do not.

While myocarditis has long been considered a reversible disease, advances in cardiac magnetic resonance imaging (MRI) have made it possible to show that myocardial damage persists long after the acute episode has been resolved. For instance, Grün et al found that, in about 50% of cases, scarring was still visible in the myocardium on MRI imaging (with late gadolinium enhancement (LGE)) 4 years after the initial event [5].

In addition, several studies have shown that LGE is a prognostic marker in myocarditis [5,6] and other non-ischemic heart disease [7,8] and that there is a strong correlation between LGE and ventricular arrhythmias in dilated [9] and hypertrophic [10] cardiomyopathies.

These findings led us to hypothesize that there is a persistent risk of severe ventricular arrhythmia at a distance from an episode of acute myocarditis associated with VT/VF.

The aim of this study was to evaluate and compare the rate of major arrhythmic events (MAEs) during follow-up in two groups: patients who experience VT/VF in the acute phase of myocarditis and patients who experience VT/VF on myocarditis sequelae.

## 2. Experimental Section

### 2.1. Study Population

This retrospective observational study was conducted in 15 French university hospitals: Besançon, Brest, Clermont-Ferrand, Dijon, Lyon (Louis Pradel), Marseille (Hôpital Nord), Nancy, Nantes, Paris (Hôpital Bichat Claude Bernard, Hôpital Européen Georges Pompidou and Hôpital de la Salpêtrière), Poitiers, Reims, Rennes and Saint-Etienne.

According to institutional policy, approval from our Institutional Review Board was not required. 

Inclusion criteria were as follows:

1. Documented sustained VT or VF.

The defibrillator’s Holter function allowed for the exhaustive collection of MAEs and associated treatments (anti-tachycardia pacing (ATP) or shock). All the data were collected retrospectively.

VT was defined as ventricular tachycardia occurring in the VT zone of the ICD and treated with either ATP or shock, depending on the ICD programming.

VF was defined as ventricular tachycardia occurring in the VF zone of the ICD, for which the patient either received ATP during ICD charge or shock.

2. Acute myocarditis or sequelae from previous myocarditis confirmed on cardiac MRI.

A diagnosis of acute myocarditis was based on the diagnosis provided in the patient’s medical file, and it was confirmed by two independent investigators (L.R./C.G.) according to the current state of knowledge of the European Society of Cardiology Working Group on Myocardial and Pericardial Diseases [11]. In case of missing data or discrepancy, the patient was not included (Figure 1).

Acute myocarditis was suspected in the presence of one or more clinical features (i.e., acute chest pain, dyspnea, palpitations and/or unexplained arrhythmia symptoms and/or syncope and/or aborted sudden cardiac death) and one or more of the diagnostic criteria from a complementary exam (i.e., newly abnormal 12 lead ECG, myocardiocytolysis markers (elevated TnT/TnI), or functional and structural abnormalities on cardiac imaging (echo/angio/cardiac MRI)).

Cardiac MRI was performed on all patients for tissue characterization: edema and/or LGE of classical myocarditic pattern. The Lake Louise criteria [12] were used when available (at least two of the following criteria: tissue edema, hyperemia and LGE).

Endomyocardial biopsy is not routine practice and was not done on any of our subjects, but cardiac MRI and coronary examination (coronary angiography or coronary computed tomography angiography) were systematically performed in order to exclude coronary artery disease.

After other heart diseases were ruled out, in particular ischemic heart disease, myocarditis sequelae were diagnosed with the presence of subepicardial LGE on MRI [13] in patients with a history of myocarditis (confirmed or suspected) and with no evidence of a clinically acute episode of myocarditis.

3. Implantation of an ICD after arrhythmia and before hospital discharge.

We excluded patients with LVEF ≤ 35% (formal ICD indication), a missing period of follow-up between the initial arrhythmic event and implantation of the ICD (hospital discharge after VT/VF without ICD or wearable defibrillator), giant cell myocarditis, sarcoidosis or other suspected causes of arrhythmia.

### 2.2. Screening Methodology

Patients in Besançon, Dijon, Clermont-Ferrand, Nantes, Poitiers, Reims and Rennes were recruited through the hospital’s Department of Medical Information (DIM) using the International Classification of Diseases (ICD-10) diagnostic codes for myocarditis and the Common Classification of Medical Procedures (Classification Commune des Actes Medicaux, CCAM) codes for the implant of a defibrillator (Appendix A). For the other centers, patients implanted with defibrillators were identified from pre-existing internal registries (Brest, Lyon, Marseille, Marseille, Nancy, Paris (Hôpital Bichat Claude Bernard, Hôpital Européen Georges Pompidou, Hôpital de la Pitié-Salpêtrière) and Saint-Etienne).

As a comparison cohort, we recruited patients who were diagnosed with sustained ventricular arrhythmia during the acute phase of myocarditis and not subsequently implanted with an ICD from two separate myocarditis registries (Hôpital de la Pitié-Salpêtrière, Paris, and Rennes University Hospital).

### 2.3. Endpoints

The primary endpoint was the occurrence of a MAE (i.e., any appropriate intervention of the defibrillator (ATP or shock, as appropriate) on tachycardia or ventricular fibrillation). An electrical storm was defined as at least three MAEs in less than 24 h.

The other collected variables included cardiovascular risk factors (age, male sex, high blood pressure, diabetes, dyslipidemia, current smoking); personal history of atrial fibrillation and heart disease and family history of sudden death; the type of initial arrhythmia more or less complicated by cardiorespiratory arrest; laboratory testing with troponin and C-reactive protein peaks during hospitalization; electrocardiogram with rhythm after treatment for tachycardia, conduction and repolarization; echocardiography (during hospital stay) with LVEF (biplane Simpson method), left ventricular end-diastolic diameter (LVED) to define a dilated left ventricle if greater than 52 mm in women and 58 mm in men [14], presence of pericardial effusion and left ventricular segment kinetics; cardiac MRI with LVEF, left ventricular end-diastolic volume and LGE; treatment at discharge from hospital (beta blockers, amiodarone, renin–angiotensin–aldosterone system inhibitors); defibrillator complications (lead repositioning, inappropriate shock, lead fracture, endocarditis, pneumothorax); possible preimplantation programmed ventricular pacing; and ventricular tachycardia ablation procedures before or after the implantation of the defibrillator.

### 2.4. Statistical Analyses 

The statistical results of the continuous variables are presented as means ± standard deviation for Gaussian distribution, medians (first quartile to third quartile) for non-Gaussian distribution after the Shapiro–Wilk test, and the results of the dichotomous variables as *n* (%). For categorical data, a chi-square or Fischer exact tests were used, while a Student’s *t*-test was used for the comparison of continuous data with normal distribution variables or a Mann–Whitney test for non-parametric variables. The threshold of significance was set at 5%. The Kaplan–Meier survival curves were constructed to study the occurrence of MAEs at follow-up and compared with the log-rank test. Data were censored at the date of the MAE episode, the last follow-up appointment or death. Variables entered into the multivariate model were chosen according to their univariate relationship with an inclusion cut-off at 20% and an exclusion cut-off at 10%. Cox multivariate stepwise backward conditional regression analysis was performed to test for predictors of MAEs in the total ICD population. The statistical tests were performed with SPSS software version 22 (IBM Corp., Armonk, N.Y., USA).

## 3. Results

### 3.1. Population Characteristics 

We included a total of 68 patients treated in 15 French university hospitals over a 10 year period (2007–2017) in the primary analysis: 28 in the acute myocarditis group and 40 in the myocarditis sequelae group (Figure 1).

Group comparisons revealed no significant differences in age, sex, cardiovascular risk factors, history of atrial fibrillation, heart disease or family history of sudden death (Table 1).

### 3.2. Initial Ventricular Arrhythmia 

Ventricular fibrillation was the most common initial ventricular arrhythmia in the acute myocarditis group (58%), and ventricular tachycardia was most common in the myocarditis sequelae group (78%); the difference in initial ventricular arrhythmia type was significant (*p* = 0.006) (Table 2). 

Cardiorespiratory arrest was a complication in 68% (*n* = 19) of patients from the acute group but only in 30% (*n* = 12) of cases from the sequelae group (*p* = 0.003).

### 3.3. Complementary Exams 

After treatment of the initial arrhythmia, supraventricular rhythm disorders were observed in 11% (8 patients, 7 AF episodes and 1 focal atrial tachycardia), wide QRS in 33% and repolarization disorder in 60% in the acute myocarditis group, compared with 8%, 33% and 62%, respectively, in the myocarditis sequelae group. The differences were not significant (Table 2).

On transthoracic echocardiography, LVEF was significantly higher in the acute myocarditis group (53% vs. 46%, *p* = 0.019), and pericardial effusion was significantly more frequent (26% vs. 0%, *p* = 0.020). Regarding left ventricular dilation and kinetic disorders, no significant differences were found between the acute myocarditis and myocarditis sequelae groups, with left ventricle dilatation in 19% and 28% of cases, respectively, and kinetic disorders observed in 41% and 53% of cases (Table 3).

On myocardial MRI in the acute myocarditis group, LVEF was 49 ± 13% and LVED was 100 ± 26 mL/m^2^. In the myocarditis sequelae group, LVEF was 47 ± 9% and LVED 96 ± 33 mL/m^2^. On MRI imaging, all patients displayed LGE (inclusion criteria), and the LGE was located in the same areas (Table 4).

### 3.4. Treatment

There was no difference in the four drug classes studied at hospital discharge (Table 2). 

No differences were found regarding the type of defibrillator implanted, the programmed therapy zones or complications at follow-up (Table 5).

### 3.5. Major Arrhythmic Events

Over a median follow-up period of 1203 (654–2465) days in the acute myocarditis group and 1131 (589–1930) days in the myocarditis sequelae group (*p* = 0.190), 11 (39%) acute patients had at least one MAE versus 24 (60%) of sequelae patients, with no significant difference in the Kaplan–Meier curve (log-rank test, *p* = 0.060) (Figure 2). 

Kaplan–Meier MAE rate estimates at one and three years of follow-up were 19% and 45% in the acute group, and 43% and 64% in the sequelae group, respectively.

The first MAE occurred after one year for 73% of the acute group compared with 29% of the myocarditis sequelae group (*p* = 0.027). There were no differences in the number of MAEs per patient or type of MAE (Table 6 and Figure 3).

A total of 143 and 90 MAEs were identified in the acute myocarditis and myocarditis sequelae groups, respectively, with no difference in the distribution of episodes (Figure 3). Twenty-one percent of VT in the myocarditis sequelae group required an ICD shock compared with only 5% in the acute group (*p* = 0.004), and there was no difference in the type of ICD treatment for electrical storms.

After univariate analysis and multivariate Cox regression analysis (Table 7), the only predictors of MAE in the total ICD population were an anterior location of LGE on MRI (HR (95% CI): 2.60 (1.28–5.59), *p* = 0.009) and an ICD indication for myocarditis sequelae (HR (95% CI): 2.88 (1.29–6.44), *p* = 0.010) (compared to an ICD indication for acute myocarditis). The use of betablockers or amiodarone at discharge was not associated with MAE occurrence in univariate analysis.

### 3.6. Acute Myocarditis without ICD 

Nine patients presenting VT/VF during the acute phase of myocarditis without an ICD implantation at hospital discharge (non-ICD group) were recruited from two registries (Hôpital de la Pitié Salpêtrière, Paris, and Rennes). 

Over a median follow-up period of 1074 (263–2839) days, MAE occurred in 7/9 patients (among them, 1 patient died subsequently to the event, 5 patients were implanted with an ICD, and 1 patient refused the ICD implantation). Kaplan–Meier MAE rate estimates at one and three years of follow-up were 44% and 72% in this group.

The only significant difference with the implanted acute myocarditis group was the type of initial arrhythmia: 89% of VT and 11% of VF in the non-ICD group (vs. 42% and 58%; *p* = 0.021). The recurrence of MAE during follow-up was not significantly different in the implanted group and the non-implanted group (log rank *p* = 0.087; Figure 4).

## 4. Discussion

The aim of this study was to evaluate the rate of major arrhythmic events in patients with severe ventricular arrhythmia in the acute phase of myocarditis. We chose to compare the acute myocarditis group to a control group with myocarditis sequelae. The control group had two advantages for this study: a formal indication for a secondary prevention defibrillator and the fact that they were very similar to the acute myocarditis group in terms of clinical characteristics and underlying heart disease.

### 4.1. Population

Our population of 68 patients was 84% male. Male predominance in myocarditis has also been observed in other studies [15], with hormonal variations in women reportedly being identified as protective [16]. Moreover, men are twice as likely to develop myocardial fibrosis following myocarditis than women [17]. Indeed, elevated testosterone levels in males may directly promote increased inflammation, fibrosis and cardiac remodeling after acute myocarditis [18].

In the acute myocarditis and myocarditis sequelae groups, only 3 and 11 patients, respectively, had previous heart disease. During follow-up, none of the other patients developed heart disease or sarcoidosis to which the arrhythmic recurrences could be attributed.

### 4.2. Initial Arrhythmia 

VF was the most frequent initial ventricular arrhythmia in the acute myocarditis group (*p* = 0.006), which is consistent with the higher percentage of cardiorespiratory arrest observed in this group (*p* = 0.003). According to Saito et al., the pathophysiology of ventricular arrhythmias in the acute phase of myocarditis is as follows: increased ventricular vulnerability with increased triggered activity, lengthened effective refractory period and decreased Kv4.2 potassium channel expression [19].

On the contrary, there were significantly more instances of ventricular tachycardia in the myocarditis sequelae group, probably reentrant or automatic monomorphous tachycardia on the myocardial scarring.

### 4.3. Recurrence of Arrhythmia

Many studies have focused on defining the predictive factors for progression to dilated cardiomyopathy, but few have focused on the risk of ventricular arrhythmia.

It is generally assumed that at a distance from the acute phase of myocarditis, the transient pro-arrhythmogenic trigger disappears, similar to ischemia in the acute phase of a myocardial infarction. However, MAEs recurred in 39% of acute myocarditis after the acute episode was resolved. The high incidence of MAEs observed in our cohort is comparable to that observed in primary prevention ICD patients by van Welsenes et al, who described a 37% incidence of appropriate therapies of over a 5-year period [20]. Recently, in 34 patients with myocarditis sequelae implanted of an ICD for primary prevention (LVEF < 35%, 1/3 of the population) and secondary prevention (2/3 of the population), Pallicek et al. showed that up to 58% of patients presented at least one episode of ventricular arrhythmia after 5 years of follow-up [21].

In our myocarditis sequelae group, 60% of patients had at least one MAE, which is significantly higher than the incidence observed in patients implanted in secondary prevention (51%) in the study mentioned above [20]. Patients for whom myocarditis scarring is identified on MRI are, therefore, at a high risk of recurrence of ventricular arrhythmias. Ventricular tachycardia could be linked to myocardial fibrosis and chronic inflammation because of a persisting autoimmune response or a viral infection [22].

Ventricular tachycardia required a shock in 21% of the myocarditis sequelae group, which was significantly higher than in the acute myocarditis group (5%, *p* = 0.004). This was despite comparable VT and VF zones in the ICD settings, which argues in favor of the difficulty in reducing these episodes by ATP.

It is interesting to note that in the acute myocarditis group, the first MAE occurred after the first 3 months in 82% of patients; these data challenges the utility of wearing a wearable defibrillator for 3 months after the initial arrhythmia as recommended by some teams [23], and call into question the theory that the risk of arrhythmia is reduced once the acute phase of myocarditis is resolved.

### 4.4. Acute Myocarditis Groups: with ICD vs. without ICD

In the patients who were not implanted with an ICD, the one-year Kaplan–Meier MAE rate estimate was 44%, which was not significantly different from the implanted group (*p* = 0.087). This observation strengthens the results of the primary analysis and minimizes both the potential overestimation of MAEs in the ICD group due to inclusion bias and the overestimation of MAEs determined by ICD therapy.

### 4.5. Risk Stratification 

Interestingly, the majority of patients from the acute myocarditis group had a preserved LVEF (*n* = 19, 70%), which favors the assumption that there was no associated heart disease, and indicates that LVEF alone should not drive the decision to implement ICD as secondary prevention in these patients. The echocardiographic LVEF was significantly higher in the acute myocarditis group, a difference not found on MRI (which is the gold standard but performed later than the echocardiography).

In line with our inclusion criteria, late gadolinium enhancement was present in all patients, primarily in the lateral segments (acute myocarditis 77% and myocarditis sequelae 90%), which is consistent with the literature [24]. No difference in the location of LGE was observed between the two groups. However, anterior LGE was found to be an independent predictor of MAE at follow-up in the whole population. This result could reflect how LGE extent and ventricular arrhythmia risk are associated in myocarditis. Recent evidence has shown that the presence of anteroseptal LGE in patients with acute myocarditis and preserved ejection fraction is a strong predictor of worse clinical outcomes [25]. Repeated MRI at follow-up to quantify myocardial healing (through LGE extent evolution and LV function parameters) after acute myocarditis could help to stratify the risk of arrhythmic events as well as myocardial dysfunction [26,27]. Further prospective studies should focus on the role of LGE extent and location for MAE prediction in patients with myocarditis.

### 4.6. Limitations

We required a reliable record of ventricular rhythmic events; therefore, only patients implanted with a defibrillator were included in the primary analysis. It is possible that only patients at risk of rhythmic recurrence were implanted; however, the decision to implant was specific to each center and was not based on a standardized approach. The basic characteristics of the acute myocarditis group do not support this bias either, with a high number of preserved LVEF in particular. Moreover, a small number of patients (*n* = 9) from two monocenter myocarditis registries were included in the acute myocarditis without ICD group. These patients displayed the same risk of arrhythmia recurrence as the patients with ICD at discharge, minimizing the risk of inclusion bias.

Though none of the patients had a histological diagnosis of myocarditis through endomyocardial biopsy, cardiac MRI was systematically performed, providing tissue characterization and, therefore, a reliable diagnosis of acute myocarditis [11]. Due to the retrospective nature of the study and the long period of inclusion, our MRI data were based on reports alone. We were not able to estimate the extent of LGE or to obtain Lake Louise criteria for every exam, but each medical file was reviewed by two investigators who confirmed the diagnosis of myocarditis according to ESC criteria using the available clinical, biological and imaging data [11].

## 5. Conclusions

Our study shows that patients presenting ventricular tachycardia or ventricular fibrillation in the acute phase of myocarditis have a high risk of recurrence of sustained ventricular arrhythmias. If confirmed by prospective studies, these new data could support the implantation of an ICD in patients presenting sustained ventricular arrhythmias in the acute phase of myocarditis. The next step will be to identify factors that can be used to predict the recurrence of arrhythmias in order to more accurately select patients eligible for defibrillators in a prospective study.

## Figures and Tables

**Figure 1 jcm-09-00848-f001:**
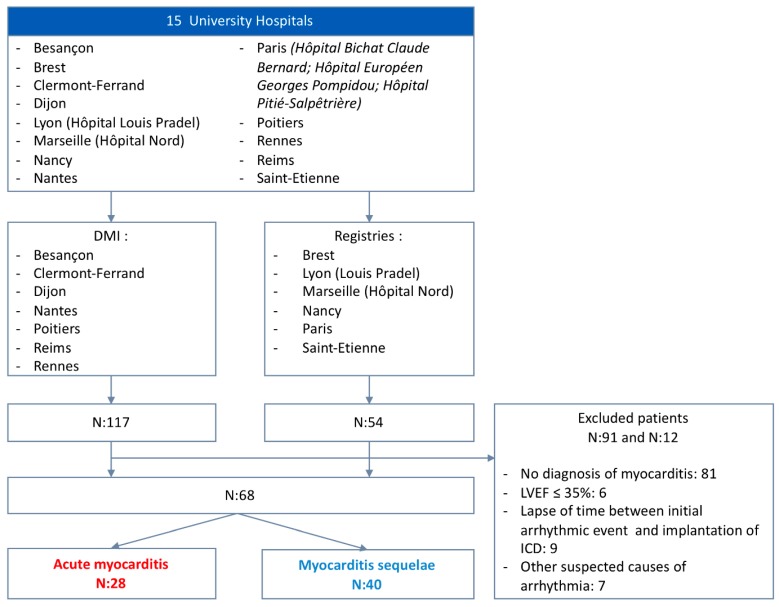
Study population.

**Figure 2 jcm-09-00848-f002:**
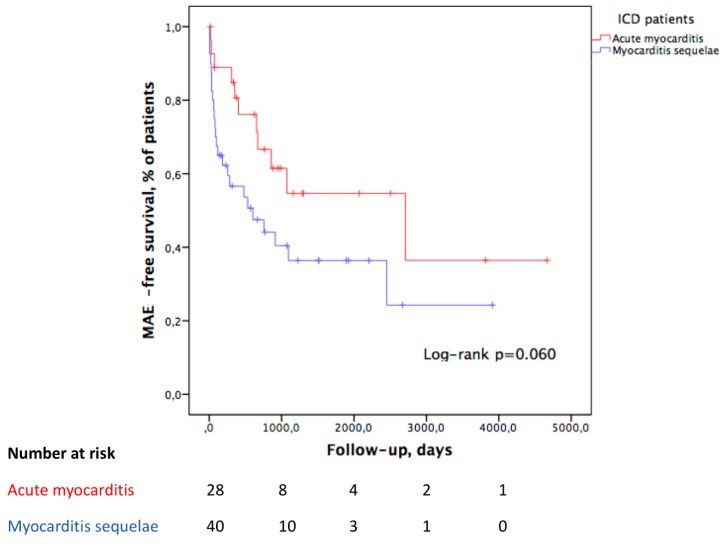
Kaplan–Meier curve comparing survival without major arrhythmic event between acute myocarditis and myocarditis sequelae patients implanted with an ICD.

**Figure 3 jcm-09-00848-f003:**
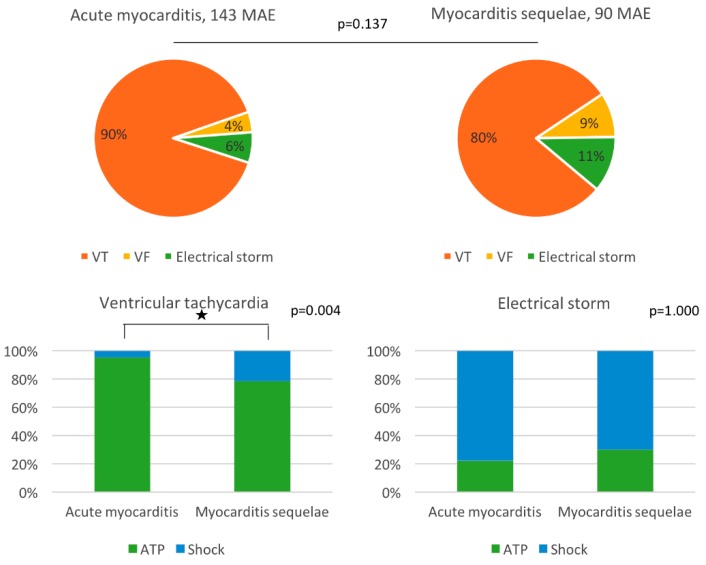
Distribution and treatment of major arrhythmic events (MAEs) in the two groups. VT: ventricular tachycardia, VF: ventricular fibrillation, ATP: antitachycardia pacing.

**Figure 4 jcm-09-00848-f004:**
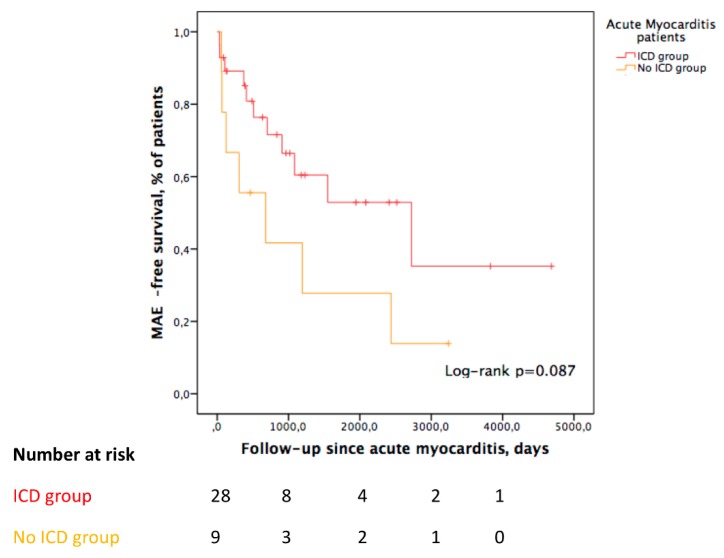
Kaplan–Meier curve comparing survival without major arrhythmic events in acute myocarditis patients with and without an ICD.

**Table 1 jcm-09-00848-t001:** Baseline characteristics.

Previous History	Acute Myocarditis (*n* = 28)	Myocarditis Sequelae (*n* = 40)	*p*
**Cardiovascular risk factors**			
Age	47 ± 18	47 ± 15	0.905
Male	21 (0.75)	36 (0.90)	0.179
High blood pressure	4 (0.14)	9 (0.22)	0.535
Diabetes	1 (0.04)	2 (0.05)	1.000
Dyslipidemia (*n* = 43)	2 (0.10)	1 (0.04)	0.607
Current smoking	3 (0.11)	5 (0.12)	1.000
**History**			
Family history of sudden death (*n* = 67)	5 (0.18)	2 (0.05)	0.120
Atrial Fibrillation (*n* = 65)	1 (0.04)	5 (0.13)	0.224
**Cardiopathy**			
Ischemic (*n* = 65)	1 (0.04)	2 (0.05)	1.000
Non-ischemic (*n* = 66)	2 (0.07)	9 (0.24)	0.100

**Table 2 jcm-09-00848-t002:** Clinical data. ACE: angiotensin-converting enzyme; ARBs: angiotensin II receptor blockers; BB: beta-blocker; CPK: creatine phosphokinase; ECG: electrocardiogram; LBBB: left-bundle branch block; RBBB: right-bundle branch block; LAFB: left anterior fascicular block; WBC: white blood cells.

Clinical Data	Acute Myocarditis (*n* = 28)	Myocarditis Sequelae (*n* = 40)	*p*
**Initial Arrhythmia Type (*n* = 61)**			0.006
Ventricular Tachycardia	10 (0.42)	29 (0.78)	
Ventricular Fibrillation	14 (0.58)	8 (0.22)	
**Cardiorespiratory arrest**	19 (0.68)	12 (0.30)	0.003
**Cardiac rhythm after resuscitation**	*n* = 27	*n* = 39	
Sinus	24 (0.89)	36 (0.92)	0.480
Atrial Fibrillation	2 (0.07)	3 (0.08)	
Atrial Tachycardia	1 (0.04)	0	
**QRS**	*n* = 27	*n* = 39	
Wide	9 (0.33)	13 (0.33)	1.000
Incomplete RBBB	5 (0.56)	3 (0.25)	
RBBB	1 (0.11)	2 (0.17)	
LBBB	0	3 (0.25)	
LAFB	3 (0.33)	2 (0.17)	
Non-specified	1 (0.11)	3 (0.25)	
**Ventricular Repolarization on the ECG**	*n* = 20	*n* = 24	
Abnormal	12 (0.60)	15 (0.62)	1.000
Negative T wave	7 (0.58)	11 (0.73)	
ST elevation	5 (0.42)	2 (0.13)	
ST suppression	4 (0.33)	3 (0.20)	
Long QT	0	1 (0.07)	
Early repolarization	2 (0.17)	1 (0.07)	
**Biological data at admission**			
Troponin (μg/L) (*n* = 41)	8 (1.9–23.3)	3.7 (1.0–43.5)	0.433
CRP (mg/L) (*n* = 39)	64 (11.2–84.5)	3 (1–9.8)	0.001
WBC count (× 10^9^/L) (*n* = 35)	17.4 (14–20.7)	11.1 (7.9–14.2)	0.001
Creatinine (mmol/L) (*n* = 48)	83 (49–123)	94 (77–127)	0.125
CPK (U/L) (*n* = 35)	411 (280–1295)	534.5 (184.8–1057.8)	0.325
**Medication at discharge**			
BB (*n* = 65)	24 (0.92)	37 (0.95)	1.000
ACE inhibitor/ARBs (*n* = 45)	14 (0.70)	16 (0.64)	0.757
Amiodarone (*n* = 63)	4 (0.15)	10 (0.27)	0.362
Aldosterone antagonist (*n* = 43)	2 (0.10)	1 (0.04)	0.590

**Table 3 jcm-09-00848-t003:** Initial Transthoracic echocardiography (TTE) data. LV: left ventricle. LVED: left ventricular end-diastolic diameter.

TTE	Acute Myocarditis	Myocarditis Sequelae	*p*
**LVEF**			
Normal (≥50%) (*n* = 67)	19 (0.70)	23 (0.57)	0.315
Value in % (*n* = 64)	53 ± 10	46 ± 11	0.019
**Pericardium**	*n* = 19	*n* = 20	
Effusion	5 (0.26)	0	0.020
**LVED**	*n* = 16	*n* = 18	
Dilated LV	3 (0.19)	5 (0.28)	0.693
**LV Kinetics**	*n* = 17	*n* = 19	
Kinetic disorders	7 (0.41)	10 (0.53)	0.525
Hypokinesis			
Global	1 (0.14)	5 (0.50)	
Inferior	4 (0.57)	7 (0.70)	
Anterior	3 (0.43)	5 (0.50)	
Septal	5 (0.71)	5 (0.50)	
Lateral	2 (0.29)	6 (0.60)	
Apical	2 (0.29)	5 (0.50)	
Dyskinesia			
Inferior	1 (0.14)	2 (0.20)	
Anterior	1 (0.14)	1 (0.10)	
Septal	1 (0.14)	0	
Lateral	1 (0.14)	1 (0.10)	
Apical	1 (0.14)	0	

**Table 4 jcm-09-00848-t004:** Myocardial magnetic resonance imaging (MRI) data. LVEF: left ventricular ejection fraction, LVEDV: left ventricular end-diastolic volume.

MRI	Acute Myocarditis	Myocarditis Sequelae	*p*
Time from arrhythmia to MRI (days) (*n* = 37)	8.5 ± 6	7.5 ± 6	0.624
LVEF (%) (*n* = 63)	49 ± 13	47 ± 9	0.479
Indexed LVEDV (ml/m^2^) (*n* = 30)	100 ± 26	96 ± 33	0.760
**Late gadolinium enhancement**	***n* = 27**	***n* = 38**	
Anterior	9 (0.33)	11 (0.29)	0.788
Lateral	20 (0.74)	32 (0.84)	0.358
Inferior	18 (0.67)	23 (0.60)	0.795
Septal	10 (0.37)	16 (0.42)	0.799
Apical	4 (0.15)	10 (0.26)	0.363

**Table 5 jcm-09-00848-t005:** Type of defibrillator implanted with ventricular tachycardia/fibrillation (VT/VF) zones and their complications BPM: beats per minute; ICD: implantable cardioverter defibrillator. 1C: single chamber. 2C: double chamber. 3C: triple chamber.

ICD	Acute Myocarditis	Myocarditis Sequelae	*p*
**Type**	***n* = 26**	***n* = 37**	0.171
1C	14 (0.54)	22 (0.59)	
2C	6 (0.23)	12 (0.32)	
3C	0	1 (0,03)	
Subcutaneous	6 (0.23)	2 (0.06)	
**Programmed therapy zones (BPM)**	***n* = 13**	***n* = 18**	
VT	181 ± 23	176 ± 17	0.503
VF	231 ± 13	223 ± 15	0.101
**Complications**			
Patients with at least 1 complication (*n* = 62)	9 (0.32)	5 (0.15)	0.132
Total number of complications	11	5	
Lead repositioning	3 (0.27)	1 (0.20)	
Inappropriate shock	3 (0.27)	4 (0.80)	
Lead fracture	3 (0.27)	0	
Endocarditis	1 (0.09)	0	
Pneumothorax	1 (0.09)	0	

**Table 6 jcm-09-00848-t006:** Description of major arrhythmic events (MAEs). ICD: implantable cardioverter defibrillator, VT: ventricular tachycardia, VF: ventricular fibrillation.

Major Arrhythmic Events	Acute Myocarditis (*n* = 11)	Myocarditis Sequelae (*n* = 24)	*p*
**Time from ICD implantation to 1st MAE (days)**			
Median	510 (240–804)	111 (44–338)	0.078
≤3 months	2 (0.18)	11 (0.46)	0.150
≥1 year	8 (0.73)	7 (0.29)	0.027
**Number of MAEs**			
Median number of MAEs	4 (2–11)	2 (1–6)	0.114
1	2 (0.18)	5 (0.23)	
2	1 (0.09)	9 (0.41)	
3	2 (0.18)	1 (0.04)	
≥4	6 (0.54)	7 (0.32)	
**Number of patients presenting:**			
≥1 VT	10 (0.91)	18 (0.75)	0.223
≥1 VF	3 (0.27)	3 (0.12)	0.352
≥1 Electrical storm	6 (0.54)	8 (0.33)	0.283

**Table 7 jcm-09-00848-t007:** Univariate and multivariate backward stepwise Cox regression analysis to estimate predictors of major arrhythmic ventricular events in the total ICD population (*n* = 61 after exclusion of 7 missing data patients).

	Univariate			Multivariate		
Variable	HR	95% CI	*p*	HR	95% CI	*p*
**Sequelae group** **(vs. acute myocarditis)**	2.26	1.04–4.91	0.041	2.88	1.29–6.44	0.010
**LVEF < 50%**	1.76	0.87–3.58	0.119			
**Wide QRS**	1.94	0.83–4.53	0.124			
**Anterior LGE location**	2.05	1.01–4.17	0.047	2.60	1.28–5.59	0.009

CI: confidence interval; HR: hazard ratio; LGE: late gadolinium enhancement; LVEF: left ventricular ejection fraction.

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
