# Peer review of "High Risk of Sustained Ventricular Arrhythmia Recurrence After Acute Myocarditis"

_jcm, 2020, doi:10.3390/jcm9030848_

Round 1
Reviewer 1 Report
The authors report arrhythmic events in patients with myocarditis. The study is interesting and the manuscript was well written.
- What are the exact diagnostic criteria for acute myocarditis and myocarditis sequelae for differentiating acute myocarditis from myocarditis sequelae?
- (Table 1) How did the authors distinguish myocarditis from cardiomyopathies?
- The authors need to discuss why ventricular fibrillation was common in acute myocarditis and why ventricular tachycardia was common in myocarditis sequelae.
- (Table 2) When did the authors measure troponin and CRP?
- When troponin level is high, it is highly likely to be diagnosed with acute myocarditis by definition that the authors described. Why were patients with a high level of troponin classified into myocarditis sequelae?
- (Table7) Why was anterior LGE associated with ventricular tachyarrhythmia?
Author Response
We thank the reviewer for his/her careful review and for his/her suggestions that will certainly improve the quality of the final version of the manuscript
What are the exact diagnostic criteria for acute myocarditis and myocarditis sequelae for differentiating acute myocarditis from myocarditis sequelae?
We apologize for the definition of sequelae that wasn’t accurate enough. We added the following sentences:
After other heart diseases were ruled out, in particular ischemic heart disease, myocarditis sequelae was diagnosed with the presence of subepicardial late gadolinium enhancement on MRI [13] in patients with a previous history of myocarditis (confirmed or suspected) and with no evidence of clinically acute myocarditis syndrome.
(Table 1) How did the authors distinguish myocarditis from cardiomyopathies?
In our study, myocarditis sequelae was defined with the presence of subepicardial late gadolinium enhancement on MRI [13] in patients with a previous history of myocarditis (confirmed or suspected) and with no evidence of clinically acute myocarditis syndrome.
Other cardiomyopathies were ruled out by cardiac MRI and coronary examination (coronary angiography or coronary computed tomography angiography) that were systematically performed, but also by the clinical context of the patient and the follow-up and genetic data, according to each center. At the end, as you can see in the Figure 1, many patients were excluded due to uncertain diagnosis, and we only kept confirmed myocarditis, acute and sequelae.
However, in the absence of EMB use in France, myocarditis diagnosis is based on an combination of clinical, biological, imaging and follow-up data, as stated in the limitation chapter: “every medical file was reviewed by two investigators to confirm the diagnosis of myocarditis according to the clinical, biological and imaging data available according to ESC criteria”
The authors need to discuss why ventricular fibrillation was common in acute myocarditis and why ventricular tachycardia was common in myocarditis sequelae.
We added the following paragraph in the discussion:
Initial arrhythmia
VF was the most frequent initial ventricular arrhythmia in the acute myocarditis group (p=0.006), which is consistent with the higher percentage of cardiorespiratory arrest observed in this group (p=0.003). According to Saito et al., the pathophysiology of ventricular arrhythmias in the acute phase of myocarditis is as follows: increased ventricular vulnerability with increased triggered activity, lengthened effective refractory period and decreased Kv4.2 potassium channel expression [18].
On the contrary, there were significantly more instances of ventricular tachycardia in the myocarditis sequelae group, probably reentrant or automatic monomorphous tachycardia on the myocardial scarring.
(Table 2) When did the authors measure troponin and CRP?
At patient admission for the initial acute ventricular arrhythmia, before the ICD implant. This data was added in Table 1.
When troponin level is high, it is highly likely to be diagnosed with acute myocarditis by definition that the authors described. Why were patients with a high level of troponin classified into myocarditis sequelae?
These patients were classified as myocarditis sequelae because the clinical event was not compatible with the Lake Louise and ESC definition of acute myocarditis, mainly due to the lack of acute clinical event compatible with myocarditis and also with clinical history of previously known myocarditis with no change in MRI LGE extent at the time of the arrhythmia. In our opinion, the troponin increase observed in these patients is likely due to cardiac arrest or fast VT.
(Table7) Why was anterior LGE associated with ventricular tachyarrhythmia?
We added the following sentence in the discussion:
Recent evidence has shown that the presence of anteroseptal LGE in patients with acute myocarditis and preserved ejection fraction is a strong predictor of worse clinical outcomes (Aquaro et al. J Am Coll Cardiol. 2017 Oct 17;70(16):1977-1987).
Reviewer 2 Report
Dear authors,
Thank you for submitting this very interesting manuscript to JCM.
Arrhythmias in myocarditis and their prevention represent a very important topic to discuss, with several grey zones and knowledge gaps. The data about MRI-LGE as predictor of MAE is definetely very interesting.
However, I believe some point of the manuscript should be reviewed.
- ABSTRACT: Please, rewrite these sentences in a more understandable way "We sought to assess the risk of major arrhythmic ventricular events (MAE) over time in patients implanted with an ICD following sustained VT/VF in the acute phase of myocarditis compared to those implanted for VT/VF occurring on myocarditis sequelae. Our retrospective 43 observational study included patients implanted with an ICD following VT/VF during acute myocarditis or VT/VF on myocarditis sequelae, from 2007 to 2017, in 15 French university hospitals. "
- INTRODUCTION: "Acute myocarditis is associated with arrhythmias in about one quarter of patients; one third of the arrhythmias are either ventricular tachycardia (VT) or ventricular fibrillation". Please, add a reference for this statement.
- INCLUSION CRITERIA: a. please define documented VT and VF; b. please, define sequalae; c. please define other suspected causes of arrhythmia.
- ENDPOINTS: a. please define cardiovascular risk factors; b. did you check Hs-troponin? if yes, please say so; c. please clarify which method did you use to calculate LVEF; d. please define defibrillator complications.
- RESULTS: a. I believe data on CRP are quite obvious, I would delete them. Did you study any other blood parameter (i.e. BNP)? b. please, specify which sort of supraventricular arrhythmia. c. did you find any difference in patients taking BB or Amiodaron?
- Did patients have MRI-compatible ICD? Please specify
- Do you believe it would be worthwhile studying LGE with MRI-based follow up?
- Do you believe MRI-LGE has any space in indications for ICD? Please, try to discuss these points in your conclusion.
Author Response
We thank the reviewer for his/her careful review and for his/her suggestions that will certainly improve the quality of the final version of the manuscript
Arrhythmias in myocarditis and their prevention represent a very important topic to discuss, with several grey zones and knowledge gaps. The data about MRI-LGE as predictor of MAE is definetely very interesting.
However, I believe some point of the manuscript should be reviewed.
ABSTRACT: Please, rewrite these sentences in a more understandable way "We sought to assess the risk of major arrhythmic ventricular events (MAE) over time in patients implanted with an ICD following sustained VT/VF in the acute phase of myocarditis compared to those implanted for VT/VF occurring on myocarditis sequelae. Our retrospective observational study included patients implanted with an ICD following VT/VF during acute myocarditis or VT/VF on myocarditis sequelae, from 2007 to 2017, in 15 French university hospitals. "
We changed the abstract as follows: The aim of this multicenter retrospective study was to evaluate and compare the rate of major arrhythmic events (MAE) during follow-up in two groups: patients who experience VT/VF in the acute phase of myocarditis and patients who experience VT/VF on myocarditis sequelae. We included patients implanted from 2007 to 2017 with an ICD in 15 French university hospitals, following VT/VF during acute myocarditis or VT/VF on myocarditis sequelae.
INTRODUCTION: "Acute myocarditis is associated with arrhythmias in about one quarter of patients; one third of the arrhythmias are either ventricular tachycardia (VT) or ventricular fibrillation". Please, add a reference for this statement.
We checked the cited reference, and it appears to be the correct reference in regard to the rates of events cited: Anzini, M.; Merlo, M.; Sabbadini, G.; Barbati, G.; Finocchiaro, G.; Pinamonti, B.; Salvi, A.; Perkan, A.; Di Lenarda, A.; Bussani, R., et al. Long-term evolution and prognostic stratification of biopsy-proven active myocarditis. Circulation 2013, 128, 2384-2394, doi:10.1161/CIRCULATIONAHA.113.003092.
However, we added a reference to a new and important paper on the subject that reviews specifically the epidemiology and pathophysiology of arrhythmias in myocarditis: Peretto, G.; Sala, S.; Rizzo, S.; De Luca, G.; Campochiaro, C.; Sartorelli, S.; Benedetti, G.; Palmisano, A.; Esposito, A.; Tresoldi, M., et al. Arrhythmias in myocarditis: State of the art. Heart Rhythm 2019, 16, 793-801, doi:10.1016/j.hrthm.2018.11.024.
INCLUSION CRITERIA: a. please define documented VT and VF; b. please, define sequalae; c. please define other suspected causes of arrhythmia.
We apologize for the confusion, all the definitions were listed below this list at the end of the chapter, and we added it after each inclusion criteria.
ENDPOINTS: a. please define cardiovascular risk factors;
We added the details: age, male sex, high blood pressure, diabetes, dyslipidemia ,current smoking;
- did you check Hs-troponin? if yes, please say so;
No, Hs troponin was not used in the centers during the inclusion period
- please clarify which method did you use to calculate LVEF
We added the details: Biplane Simpson method
; d. please define defibrillator complications.
We added the following details: lead repositioning, inappropriate shock, lead fracture, endocarditis, pneumothorax
RESULTS: a. I believe data on CRP are quite obvious, I would delete them.
The sentence about CRP has been removed
Did you study any other blood parameter (i.e. BNP)?
We also collected CPK release, creatinine value and leucocyte count, and we added this information in Table 2
- please, specify which sort of supraventricular arrhythmia.
We added the details in the results (8 patients, 7 AF episodes and 1 focal atrial tachycardia)
- did you find any difference in patients taking BB or Amiodaron?
We suppose that the reviewer is asking about the difference between SVT and VT episodes with regard to discharge treatments? No significant difference was observed regarding BB and amiodarone use regarding VT/VF recurrences at follow-up, and neither with SVT episodes. However, most patients were treated with betablockers, thus leading to a lack of statistical power in the analysis of patients without bb.
For betablockers at discharge, the Cox univariate analysis odd ratio for MAE at follow up was 1.53 (0.37-6.64), p =0.54
For amiodarone at discharge, the Cox univariate analysis odd ratio for MAE at follow up was 0.89 (0.36-2.21), p =0.81
We added the following sentence in the results:
The use of betablockers or amiodarone at discharge was not associated with MAE occurrence in univariate analysis.
Did patients have MRI-compatible ICD? Please specify
Some of the patients (mainly the most recently implanted ones) were equipped with MRI-compatible ICD, but we did not systematically collect this data, so we are unable to give the precise rate of MRI compatible ICD in our population. However, no repeated cMRI was performed at follow-up, even in MRI compatible ICD patients in our cohort.
Do you believe it would be worthwhile studying LGE with MRI-based follow up?
We thank the reviewer for this relevant comment, indeed we are currently performing a prospective study in our center with repeated MRI in ACS-like myocarditis patients follow-up at 3 months and one year. We added a sentence about this study and the abstracts references that were presented at the ESC recently in the discussion.
Repeated MRI at follow-up to quantify myocardial healing (through LGE extent evolution and LV function parameters) after acute myocarditis could help to stratify the risk of arrhythmic events as well as myocardial dysfunction[25,26].
Do you believe MRI-LGE has any space in indications for ICD? Please, try to discuss these points in your conclusion.
The reviewer raises a very good point - if LVEF is not able to risk stratify the myocarditis patients (with LVEF >35% at implantation in our study), what are the risk markers that could be used ? In our study, the anterior LGE location was associated with MAE. It is possible that the extent of LGE (usually, anterior myocarditis is more extended than lateral or inferior myocarditis) could be used as a predictor of MAE. However, given the very limited number of patients in the particular setting of acute myocarditis complicated from a cardiac arrest on VT/VF, a prospective study would have to be international in order to obtain a sufficient number of patients to conclude on the role of MRI. Moreover, in the near future we aim to gather all the MRI data from our patients in order to try to correlate LGE extent to MAE occurrence. But the main goal of the present study was to demonstrate that even acute myocarditis patients have a high risk of MAE at follow-up and that they should at least be very carefully monitored, and also maybe receive an ICD.
We added the following sentences in the discussion:
Recent evidence has shown that the presence of anteroseptal LGE in patients with acute myocarditis and preserved ejection fraction is a strong predictor of worse clinical outcomes (Aquaro et al. J Am Coll Cardiol. 2017 Oct 17;70(16):1977-1987).
Further prospective studies should focus on the role of LGE extent and location for MAE prediction in patients with myocarditis.